



The Impact of Climate on Surging at Donjek Glacier, Yukon, Canada
William Kochtitzky[1,2], Dominic Winski[1,2], Erin McConnell[1,2], Karl Kreutz[1,2], Seth Campbell[1,2],
Ellyn M. Enderlin[1,2,3], Luke Copland[4], Scott Williamson[4], Brittany Main[4], Christine Dow[5],
Hester Jiskoot[6]
*[1]School of Earth and Climate Sciences, University of Maine, Orono, Maine, USA*
*[2]Climate Change Institute, University of Maine, Orono, Maine, USA*
*[3]Department of Geosciences, Boise State University, Boise, Idaho, USA*
*[4]Department of Geography, Environment and Geomatics, University of Ottawa, Ottawa, ON,*
*Canada*
*[5]Department of Geography and Environmental Management, University of Waterloo, Waterloo,*
*ON, Canada*
*[6]Department of Geography, University of Lethbridge, Lethbridge, AB, Canada*
*Correspondence to:* William Kochtitzky (william.kochtitzky@maine.edu)
**Abstract.** Links between climate and glacier surges are not well understood, but are required to
enable prediction of glacier surges and mitigation of associated hazards. Here, we investigate the
role of snow accumulation and temperature on surge periodicity, glacier area changes, and
timing of surge initiation since the 1930s for Donjek Glacier, Yukon, Canada. Snow
accumulation measured in three ice cores collected at Eclipse Icefield, at the head of the glacier,
indicate that a cumulative accumulation of 13.1-17.7 m w.e. of snow occurred in the 10-12 years
between each of its last eight surges. This suggests that a cumulative accumulation threshold
must be passed before the initiation of a surge event, although it remains unclear whether the
relationship between cumulative snowfall and surging is due to the consistency in repeat surge
interval and decadal average precipitation, or if it is indeed a prerequisite to surging. We also
examined the 1968 to 2017 climate record from Burwash Landing, 30 km from the glacier, to
determine whether a relationship exists between surge periodicity and an increase of 2.5°C in
mean annual air temperature over this period. No such relationship was found, although each of
the past 8 surge events has been less extensive than the previous, with the terminus area
approximately 7.96 km$^2$ smaller after the 2012-2014 surge event compared to the ~1947 surge
event. This study shows that the impacts of climate and surging is not yet understood and
suggests that internal glacier processes may play a more important role in controlling glacier
surge events.

**1.**     **Introduction**



Surge-type glaciers account for ~1% of glaciers globally (Sevestre and Benn, 2015), but can be
the dominant glacier type in some regions (e.g., Clarke et al., 1986; Jiskoot et al., 2003), and are
important for understanding ice flow instabilities and anomalous glacier response to climate
change (Yde and Paasche, 2010). Surge-type glaciers have long periods of flow at rates below
their balance velocity (quiescent phase), typically on the order of decades, which are interrupted
by short-lived phases of glacier flow at rates much higher than the balance velocity (active phase
or surge phase), typically on the order of months to years, that are driven by internal instabilities
and sometimes lead to a marked frontal advance (Meier and Post, 1969; Clarke, 1987). When a
glacier surges, its reservoir zone at higher elevations loses mass and its receiving zone at lower
elevations gains mass, with the line of zero net mass change defined as the dynamic balance line
(DBL: Dolgoushin and Osipova, 1975). When mass gain in the receiving zone leads to a
significant advance of the terminus, an increased calving flux or other proglacial hazards can
occur.
Surges of temperate glaciers are typically hypothesized to initiate when a critical basal
shear stress is reached in a surge initiation region, causing the subglacial hydrologic system to
reorganize and the glacier to rapidly redistribute its accumulated mass down-glacier (Meier and
Post, 1969; Raymond, 1987; Eisen et al., 2005). While this hydrologic mechanism dominates
Yukon-Alaska type surging, a thermal triggering mechanism (i.e., surging controlled by basal ice
temperature), or combined hydro-thermodynamic mechanism, has been documented in surges of
polar and polythermal glaciers, such as those in Svalbard and smaller glaciers in Yukon-Alaska
(Murray et al., 2003; Frappé and Clarke, 2007; De Paoli and Flowers, 2009; Dunse et al., 2015).
Finally, overarching theories related to balance flux (Budd, 1975) and enthalpy (Sevestre et al.,
2015) have been proposed as well.
The length of a surge cycle (i.e., combined quiescent and active phases) is typically
consistent for an individual glacier, and is proportional to the length of the surge phase (Meier
and Post, 1969; Dowdeswell and others, 1991). In turn, quiescence duration is controlled by
mass balance conditions (Robin and Weertman, 1973), meaning that surge periodicity is
inversely related to accumulation rates (Dyurgerov *et al.*, 1985; Osipova and Tsvetkov, 1991;
Dowdeswell *et al.*, 1991). Prolonged quiescent phases (decades to centuries) typical of the
Svalbard region have been ascribed to low accumulation rates, often only on the order of 0.3-0.6
m a$^{-1}$ (Dowdeswell *et al.*, 1995), while short repeat intervals (12-20 years) on Variegated Glacier,



AK, correspond to accumulation rates on the order of 1.4 m a$^{-1}$ (Eisen et al., 2001; Van Geffen
and Oerlemans, 2017). However, there can be large variations in surge periodicity between
glaciers in the same region. For example, Icelandic glaciers have irregular quiescent intervals, 5-
30 years for some glaciers and up to 100-140 for others (Björnsson et al., 2003: Sigurdsson,

2005).

Changes in surge recurrence interval has been linked to changing cumulative mass

balance (Dowdeswell et al., 1995; Copland et al., 2011; Eisen et al., 2001; Striberger et al.,
2011). Dowdeswell et al. (1995) found a persistent negative mass balance reduced the glacier
surge activity in Svalbard. Conversely, an increase in precipitation and positive glacier mass
balance on Karakoram glaciers is associated with an elevated number of surge events, although it
is unclear whether the increase in accumulation (Copland et al., 2011) or increase in intense
short-term melting periods Hewitt (2007) drove the increase in surging. Eisen et al. (2001)
reported a variable surge recurrence interval that was consistent with changing amounts of
precipitation on Variegated Glacier, Alaska. Similarly, Striberger et al. (2011) found a variable
surge repeat interval at Eyjabakkajökull, Iceland associated with changes in climatically-driven
mass balance.

Previous efforts to examine connections between cumulative snow accumulation and

length of the quiescent phase have used mass balance models, off-ice meteorological
measurements, and a limited record of in situ mass balance measurements (Eisen et al., 2001;
Tangborn, 2013; Dyurgerov et al., 1985). Although these studies found that a snow accumulation
threshold had to be reached before each surge started, this potential linkage has not yet been
tested with observations of glacier surface mass balance. Here, we use the well-documented
history of surge events at Donjek Glacier (Abe et al., 2016; Kochtitzky et al., In Review; Fig. 1),
and ice cores extracted from Eclipse Icefield at the head of the glacier (Wake et al., 2002; Yalcin
et al., 2006; Kelsey et al., 2012), to explore linkages between snow accumulation and surging
since the 1930s. We combine these observations with weather station records, digital elevation
models, and remote sensing analysis to examine the impacts of climate and ice kinematics on
surge behavior.  The combination of data from eight surge events and three independent ice core
records in the accumulation zone, make Donjek Glacier an ideal site to test the influence of
climate on surge behavior.



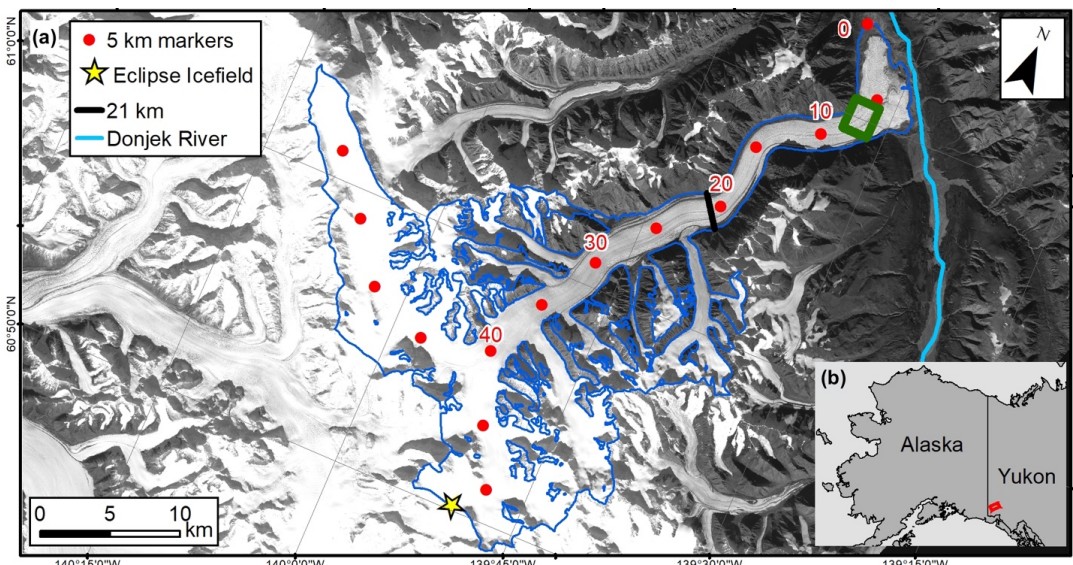

Figure 1. (a) Donjek Glacier (blue outline; RGI Consortium, 2017), with Eclipse Icefield marked

with the yellow star and Donjek River in light blue. Black line indicates the separation between

the downglacier surge-type and upglacier non-surge-type portions of the glacier. Green box

indicates extent of Figure 7a. (b) Location of Donjek Glacier in southwestern Yukon; red box

indicates extent of a. Base image from Landsat 8, 23 September, 2017.

**2.    Study Site**

Donjek Glacier (61°11'N, 139°31'W; Figure 1) is a surge-type glacier located in southwest

Yukon in the St. Elias Mountains. In 2010, Donjek Glacier was 65 km long with a surface area

of 448 km$^2$ (RGI Consortium, 2017). While the Tlingit indigenous peoples of the Yukon were

the first to observe Donjek Glacier surge (Cruikshank, 1981), the first scientific records are from

1937 in the form of Bradford Washburn's air photos (https://library.uaf.edu/washburn/).

Subsequent scientific work focused on the moraines and geomorphology (Denton and Stuvier,

1966; Johnson, 1972a and b), meteorological measurements at Eclipse Icefield as part of the

Icefield Ranges Research Project (Ragle, 1972), and surge-related outburst floods in the Donjek

River (Figure 1; Clarke and Mathews, 1981). Ice coring campaigns have occurred at least four

times at Eclipse Icefield since the 1990s and provide a wealth of snow accumulation and

atmospheric information (Wake et al., 2002; Yalcin et al., 2006; Kelsey et al., 2012).



Between May 2000 and May 2012, the area-averaged mass balance of Donjek Glacier

was -0.29 m water equivalent (w.e.) yr$^{-1}$, or -0.13 Gt yr$^{-1}$ (Larsen et al., 2015). Despite this
negative mass balance, the glacier has continued its history of frequent surging, which has
occurred approximately every 10-12 years since the 1930s (Abe et al., 2016; Kochtitzky et al., In
Review; Figure 2). Air photo records, satellite imagery and previous reports indicate that the
glacier surged in ~1935, ~1947, late-1950s, ~1969, 1977-1980, 1988-1990, 2000-2002, and
2012-2014, with progressively less extensive terminus advances up to the present day. Ice flow
velocities are only available for the two most recent surges (Abe et al., 2016; Kochtitzky et al., In
Review). Only the lower 21 km of the glacier was involved in these surge events, coinciding
with the portion of the glacier down-glacier of a valley constriction (Kochtitzky et al., In
Review; Figure 1).

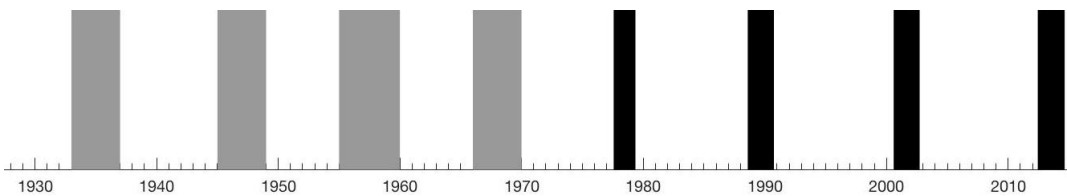


Figure 2. Surge event timing. Grey bars indicate uncertainty for surges before the satellite era.
Black bars indicate duration of active surge phase for the last four surge events, constrained by
satellite imagery.

**3.          Methods**
**3.1. Ice cores and snow accumulation record**
Ice cores were collected at Eclipse Icefield (Fig. 1) in 1996 (160 m absolute length; Yalcin and
Wake, 2001), 2002 (350 m absolute length; Fisher et al., 2004; Kelsey et al., 2012), and 2016 (59
m absolute length), to develop an understanding of past climate in the St. Elias Range. Cores
were collected during late spring, but preceding the melt season, in 1 m segments using a 8.2 cm
diameter electromechanical drill. The accumulation record from the 1996 ice core was originally
reported by Yalcin and Wake (2001) and we use their original data here. An original depth-age
scale for the 2002 core was developed by Yalcin et al. (2007).  Since then, advances in
glaciochemical signal detection, automated layer counting (Winstrup et al., 2012), and ice flow
modeling have been developed for alpine ice cores (Campbell et al., 2013; Winski et al., 2017).
We therefore applied these techniques to the existing Eclipse 2002 core data to develop updated



accumulation rate time series. The 2016 core was primarily dated using oxygen ($\delta^{15}O$) and
deuterium ($\delta D$) isotope ratios, and deuterium excess ($d_{xs}$; equation 1; Daansgaard, 1964), with
additional constraints from major ions ($Na^+$, $SO_4^{2-}$, and $Mg^{2+}$). We do not apply any thinning
corrections to the 2016 core, as it only covers the top 59 m of the firn zone and firn/ice transition
where thinning is negligible. The 2002 core was dated via annual layer counting of $\delta D$, sodium,
magnesium, calcium, and sulfate. The 2002 core was additionally constrained by known volcanic
eruption markers indicated by a spike in sulfate concentrations (Yalcin et al., 2007) and the Cs-
137 peak in 1963 from above ground nuclear testing. The seasonal timing of each of these peaks
is well characterized from previous studies in the North Pacific region (Yalcin et al. 2001, Wake
et al. 2002, Yasunari et al. 2007, Osterberg et al. 2014, Winski et al. 2017).
$$d_{xs} = \delta D - 8 \times \delta^{15}O \qquad\qquad\qquad (1)$$

Five individuals independently picked the approximate position of the 1 January marker

throughout the last 500 years for the 2002 ice core (Fig. 3a). These individual annual pick
positions were reconciled using the methods described in Winski et al. (2017), which included
incorporation of algorithum-based computer counting software (Winstrup et al., 2018). With the
resulting annually-dated timescale, annual layer thicknesses were calculated as the distance
between successive years, and water equivalent annual layer thicknesses were calculated as the
annual layer thickness multiplied by the density at the corresponding depth in the ice core. The
density for each layer was extrapolated from the 1 m-increment field density measurements.

We accounted for thinning due to glacier flow in the 2002 record using three 1-

dimensional glacier flow models, which we refer to as the Nye (Nye, 1963), Hooke (Kaspari et
al. 2008), and Dansgaard-Johnsen (Dansgaard and Johnsen, 1969) models (Fig. 3b). Following
Winski et al. (2017), we tested all reasonable combinations of free parameters in each model to
assess which model run most closely matches our observed depth-age scale (Fig. 3a). In each
model, we generated a suite of different age scales using long-term average accumulation rates
ranging from 20 to 300 cm in 10 cm increments. In the Hooke model, we also permitted the flow
parameter (*m* in Kaspari et al. 2008) to vary between 1 and 2. In the Dansgaard-Johnsen model
we permitted a flow regime change occurring between 10 and 250 m above the bed. These
activities resulted in a total of 1363 separate model runs (29 Nye, 609 Hooke and 725
Dansgaard-Johnsen), each producing a unique depth age scale.

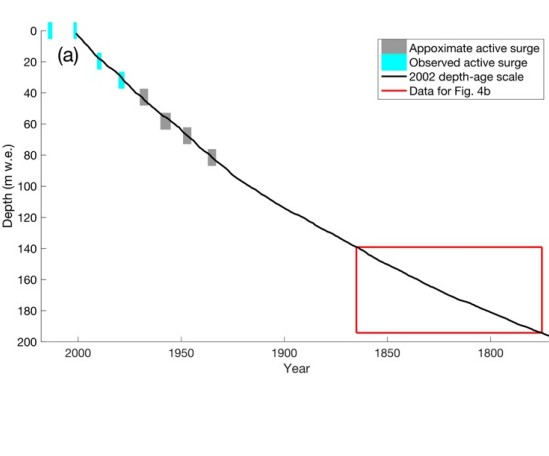

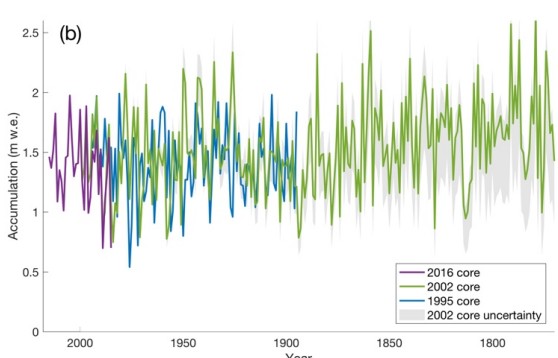


Figure 3. Ice core accumulation and depth-age scale. (a) The mean picked observed depth-age
scale from the 2002 ice core is shown in black with grey lines indicating each of the individual
picks. (b) The 1995 ice core (purple), 2002 ice core (green), and 2016 ice core (blue)
accumulation records are shown since 1770. The Nye and Hooke models are the bounds on the
2002 ice core uncertainty (grey) for accumulation from the 2002 ice core.

For each modeled depth age scale, we calculated the sum of root-mean squared errors

(RMSE) between the layer-counted and modeled depth-age scale positions at each year. Of all
combinations of models and input parameters, we found the optimized version of the Dansgaard-
Johnsen Model to produce the lowest RMSE and closest match to our observed depth-age scale.
In our modeled depth-age scale, we used 337 m w.e. (approximately 376 m of absolute
thickness) which yielded the lowest error between the optimized Dansgaard-Johnsen model (the



closest fit) and the layer counted timescale. The accumulation rate used herein is equal to the
ratio of the observed layer thickness (from the annual layer counting) over the modeled layer
thickness (from the optimized Dansgaard-Johnsen model) multiplied by 1.4 meters, which is the
optimized value of long-term accumulation that produces the best fit to the timescale. Based on
the range of results among the three flow models, the accumulation uncertainty was estimated as
±15% in the 1930s, with lower uncertainties near the top of the record.

We define our cumulative accumulation interval for each quiescent phase to stretch from

the year following surge initiation to the initiation year of the next surge (Eisen et al., 2001),
which equates to 1935-1944, 1945-1955, 1956-1966, 1967-1977, 1978-1988, 1989-2000, and
2001-2012 (Fig. 2). The surge initiation dates we use are from Kochtitzky et al. (In Review),
which are well constrained in the satellite era. Before the satellite era, our initiation years are
within the uncertainty bounds determined by Kochtitzky et al. (In Review) from advanced
terminus positions and/or push moraines captured in air photographs.

**3.2. Glacier surface elevation mapping**
Digital Elevation Models (DEMs) for 2002, 2007, 2012, and 2016 were created or obtained from
Operation IceBridge (OIB) LiDAR measurements, Satellite Pour l'Observation de la Terre 5
(SPOT-5), WorldView and the Advanced Spaceborne Thermal Emission and Reflection
Radiometer (ASTER; Table 1). OIB LiDAR tracks from 2012 and 2016 were downloaded from
the National Snow and Ice Data Center (https://nsidc.org/icebridge/portal) and down-sampled to
8 m spatial resolution for comparison with the DEMs. We obtained a 13 September 2007 SPOT-
5 DEM (40 m spatial resolution) from the SPIRIT Project (https://theia-landsat.cnes.fr) with an
uncertainty of ± 6 m (Korona and others, 2009). We received DEMs at 8 m spatial resolution
derived from WorldView imagery from the University of Minnesota Polar Geospatial Center
(PGC), with ~0.2 m vertical accuracy (Shean and others, 2016). We mosaicked the individual
WorldView DEMs from 10 August and 27 September 2013 (hereby referred to as the
August/September 2013 DEM) to create a more spatially extensive DEM of the glacier. These
2013 DEM strips do not overlap or intersect, so we are unable to quantify the potential aliasing
of glacier flow and/or melt in the mosaicked DEM. Finally, we created one 2002 DEM from
ASTER imagery using MMASTER from Girod and others (2017). The ASTER DEM has 30 m
spatial resolution and 10 m vertical uncertainty (Girod and others, 2017). We co-registered all



DEMs to the WorldView DEMs following methods from Nuth and Kääb (2011) and smoothed
extracted centerline elevation values using a 300 m moving window to visualize the data.
Table 1. Elevation data sources for ice surface change

| Source | Date | Vertical uncertainty |
|---|---|---|
| ASTER (satellite) | 26/05/2002 | 10 m |
| SPOT-5 (satellite) | 13/09/ 2007 | 6 m |
| Operation IceBridge (airborne LiDAR) | 22/05/2012 15/05/2016 | <10 cm |
| PGC/WorldView (satellite) | 10/08/2013 27/09/2013 | ~0.2 m |


### 3.3. Snowline measurements

To infer the position of the equilibrium line altitude, we manually digitized the position of the
snowline using the Landsat archive. All available cloud-free Landsat images of Donjek Glacier
were downloaded from Earth Explorer (https://earthexplorer.usgs.gov), and the last available
image of the ablation season (July, August, or September) of each year was selected to determine
the snowline for most years from 1972-2017. We additionally used one air photo from 8 July
1951, which we georeferenced with 8 tie points to produce an estimated horizontal uncertainty of
72.4 m.

We estimated the mean elevation of the snowline for each year using a 2013 WorldView

DEM (see section 3.4). We are unable to account for glacier surface elevation change over time
due to a lack of high-quality surface DEMs prior to 2002, but little change (less than 30 m) in
exposed rock along the glacier margins since the 1970s suggests that elevation changes have not
been large.

### 3.4 Ice thickness measurements

During a July, 2018 field campaign we measured ice thickness by walking with a ground
penetrating radar from Blue System Integration Ltd. (http://www.radar.bluesystem.ca/) with 5
and 10 MHz antennas to measure ice thickness over the lower ablation area of Donjek Glacier in

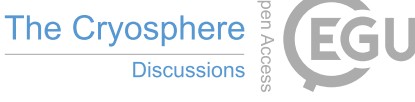


July 2018. Data were processed using IceRadarAnalyzer 4.2.5, assuming a radio-wave velocity
of 0.300 m ns$^{-1}$ in air and 0.170 m ns$^{-1}$ in ice (Mingo and Flowers, 2010).

**3.5. Climate and weather observations**
To infer climate conditions at Donjek Glacier, we use temperature and precipitation data from
the Environment and Climate Change Canada weather station at Burwash Landing (61°22'14"N,
139°2'24"W, 806 m a.s.l.), 30 km northeast of the current glacier terminus (~1000 m a.s.l.). Data
were downloaded from http://climate.weather.gc.ca using the Canadian Climate Data Scraping
Tool (Bonifacio et al., 2015). The Burwash Landing weather station has been operational since
1968 and has a nearly continuous hourly and daily record. We do not apply an elevation
correction to any weather data from Burwash Landing since we are using these data to infer
relative changes over time.
We constructed a continuous annual mean temperature record from monthly average
temperatures recorded at the weather station to examine long-term temperature change. We also
reconstructed a record of annual positive degree days (PDD) from the daily temperature data
from Burwash Landing (e.g., Ohmura, 2001). Of the 18,263 day record from 1 January 1968 to
31 December 2017, 1038 days did not have mean daily temperature readings. To fill these gaps,
we linearly interpolated missing data using the daily mean temperature observation nearest in
time. We then calculated the number of annual positive degree days by summing the daily mean
temperature for all days that exceeded 0°C for each calendar year.
We summed daily rainfall data from Burwash Landing to calculate annual liquid
precipitation. 2479 days of daily data are missing, of which 1010 occur between May 1 and
September 30, but we do not attempt to fill these, so annual estimates should be considered as
minima and are biased based on when observations occurred. The precipitation data cover
October 1966 to January 2013. These data allow us to examine the impacts of cumulative and
extreme rain events.
To complement the Burwash Landing station observations and provide a temperature
record on Donjek Glacier, we use the North American Regional Re-analysis (NARR) data set
produced by the National Center for Environmental Prediction (NCEP). NARR air temperatures
are produced through the combination of surface, radiosonde, and satellite temperature data with
the Eta atmospheric model (Mesinger et al., 2006). The surface air temperatures at three-hour



intervals for June, July and August over the period 2000 to 2016, were downscaled using
atmospheric temperatures from 16 pressure levels between 1000 hPa and 500 hPa (Jarosch et
al.,2012). The NARR air temperatures were downscaled to a resolution of 200 m, from the native
32 km, using a three-part, linear piece-wise fitting to vertical air profiles and interpolation of the
fitted parameters. The downscaled NARR air temperatures perform as well in the Yukon St.
Elias region as over British Columbia, which has been confirmed with meteorological air
temperature measurements and MODIS Land Surface Temperature measurements over regions
of permanent snow and ice (Williamson et al., 2017, Williamson et al., 2018). The 200 m
downscale produced a mean bias of 0.5°C and a mean absolute error ≤ 2°C for monthly averages
compared to 78 stations in southern British Columbia for data between 1990 and 2008 (Jarosch
et al., 2012).  We produced daily averages from the 3-hourly downscaled NARR product and
subset the data into 200 m elevation bins for Donjek Glacier using the Randolph Glacier
Inventory glacier outline. To calculate the PDDs we summed the daily averages higher than 0°C
for May through September between 1979 and 2016.

**4.    Results**
**4.1 Cumulative accumulation**
Using the cumulative annual snow accumulation from the three ice cores, we find that between
13.1 and 17.7 m w.e. (mean of 15.5 ± 1.46 m w.e.) accumulated at Eclipse Icefield between each
of the eight recent surges of Donjek Glacier (Figure 4a). While the three ice cores do not record
the same amount of accumulation each year, they also do not show a pattern of consistent spatial
bias of snow accumulation across Eclipse Icefield when compared to each other (Fig. 3).





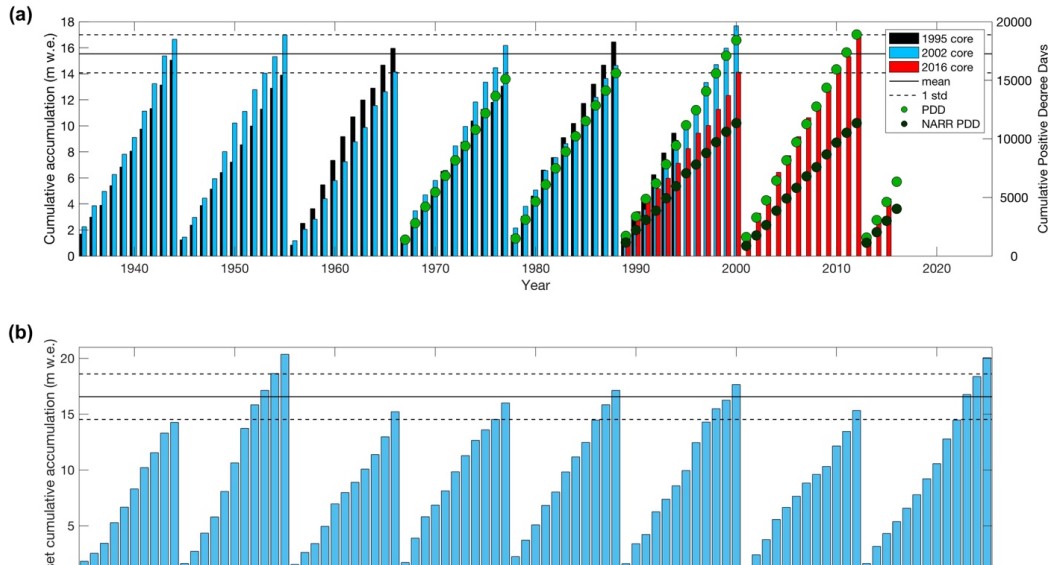


Figure 4. Cumulative accumulation between surge events. (a) Cumulative annual accumulation
from 1995 (black), 2002 (blue), and 2016 (red) ice cores between each surge event. Green
(Burwash Landing) and black (NARR) circles indicate cumulative positive degree days between
surge events on the right y-axis. (b) The cumulative accumulation from the 2002 ice core offset
by 160 years, the time need for surface snow/firn/ice to travel from Eclipse Icefield to the
constriction at 21 km from the terminus, where the surge-type portion of the glacier begins. Solid
and dashed black lines show the mean and one standard deviation cumulative accumulation
average between surge events.

Surging is limited to the lower 21 km of Donjek Glacier (Kochtitzky et al., In Review;

Figure 1), with the upper boundary of the reservoir zone coinciding with a valley constriction.
Therefore, snow accumulation 32.3 km upstream of the constriction may not have a strong
influence on surge behavior. We therefore calculated the time that it would take mass to advect
from Eclipse Icefield to the constriction from the surface flow speed, neglecting ablation and any
submergence or emergence velocity.  In 2007, the average surface flow speed was 201.4 m a$^{-1}$
along the entire length of Donjek Glacier's center flowline, with a spatial variability of 11.4 –
398 m yr$^{-1}$ over the 32.3 km trajectory between the ice cores and the constriction (Van Wychen
et al., 2018). Thus, snow that accumulates on Eclipse Icefield takes ~160 years to reach the



constriction, assuming that present-day velocities are similar to those of the past. We therefore
offset the accumulation record derived from the 2002 ice core by 160 years to reconstruct the
accumulation history preceding the surges (e.g. accumulation that reached the constriction in
2002 fell as snow in 1842; Figure 3). Using this offset record, the cumulative accumulation
between the eight surge events ranges between 14.2 and 20.4 m w.e. (mean of $16.6 \pm 2.0$ m w.e;
Figure 4b). Although this results in only a marginally wider range than the recent accumulation
history, the average cumulative accumulation is 6% lower.

**4.2 Changes in the reservoir zone surface height**
Donjek Glacier can be divided into two parts: surge-type and non-surge-type (Kochtitzky et al.,
In Review). The surge-type portion can further be divided into a reservoir zone (8-21 km
upstream of terminus) and a receiving zone (lower 8 km). The area separating the reservoir and
receiving zones, which is the area with zero net mass change during a surge event, is known as
the dynamic balance line (DBL: Dolgoushin and Osipova, 1975). Our surface DEM analysis
demonstrates that surface elevation increases in the reservoir zone following a surge event, even
though the entire reservoir zone is located in Donjek Glacier's ablation area. Between 2002 and
2007, after the 2000-2002 surge, we measured a glacier surface height increase of up to 41.6
±11.6 m in the 8-21 km reservoir zone, with an average height increase of 12.5 m (Figure 5).
From 2007-2012, covering the beginning of the 2012-2014 surge, the reservoir zone had an
average surface elevation increase of 1.0 m (Figure 5). During the surge event, the reservoir zone
decreased in surface elevation (Kochtitzky et al., In Review). From 2013-2016, a period which
includes the end of the 2012-2014 surge event, we measured an average surface elevation
increase of 10.7 m in the reservoir zone (Figure 5). After both the 2000-2002 and 2012-2014
surge events, we see refilling of the reservoir zone within five years (Figure 5).





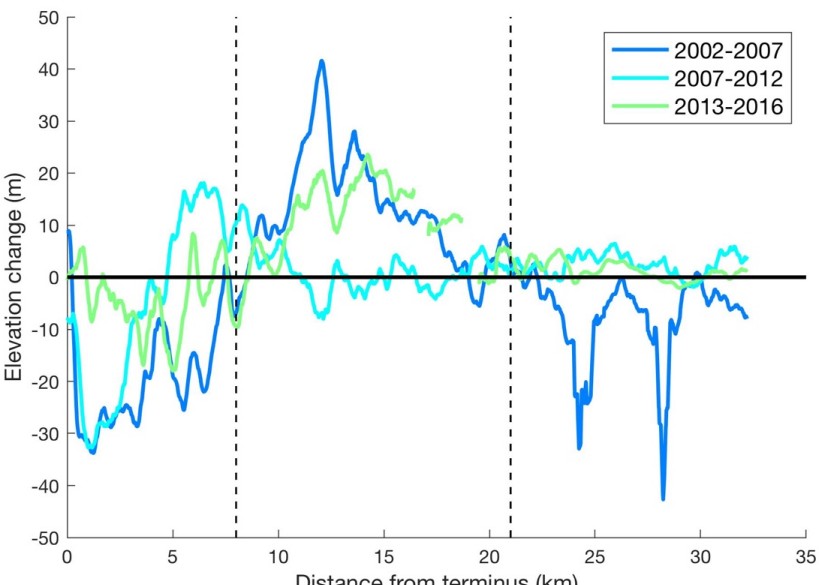


Figure 5. Surface elevation change in the reservoir zone. Surface elevation change from 2002 to
2007 (dark blue), 2007 to 2012 (light blue), and 2013 to 2016 (light green). Extent of the
reservoir zone indicated by black dashed lines at 8 km (dynamic balance line) and 21 km
(constriction) from the terminus.

**4.3 Snowline change**
Our remote sensing analysis illustrates that the summer snowline in the center flow unit of
Donjek Glacier has migrated up-glacier by 55 m yr$^{-1}$ horizontally and risen by ~1.0 m yr$^{-1}$ in
elevation over the period 1951 to 2017 (Figures 6 and 7a). Over the study period the snowline
was lowest in 1977 (Figure 7a), with an accumulation area of 337.3 km$^2$ and an Accumulation
Area Ratio (AAR) of 75.3%. The snowline reached its highest average elevation of ~2550 m
a.s.l. in 2017, corresponding to an AAR of 68.4%. Even though some snowline measurements
were made early in the ablation season, we do not find our snowline measurements to be biased
by timing of the observation, as snowline elevations in the late melt season were not consistently
different from those early in the melt season (Figure 7a).





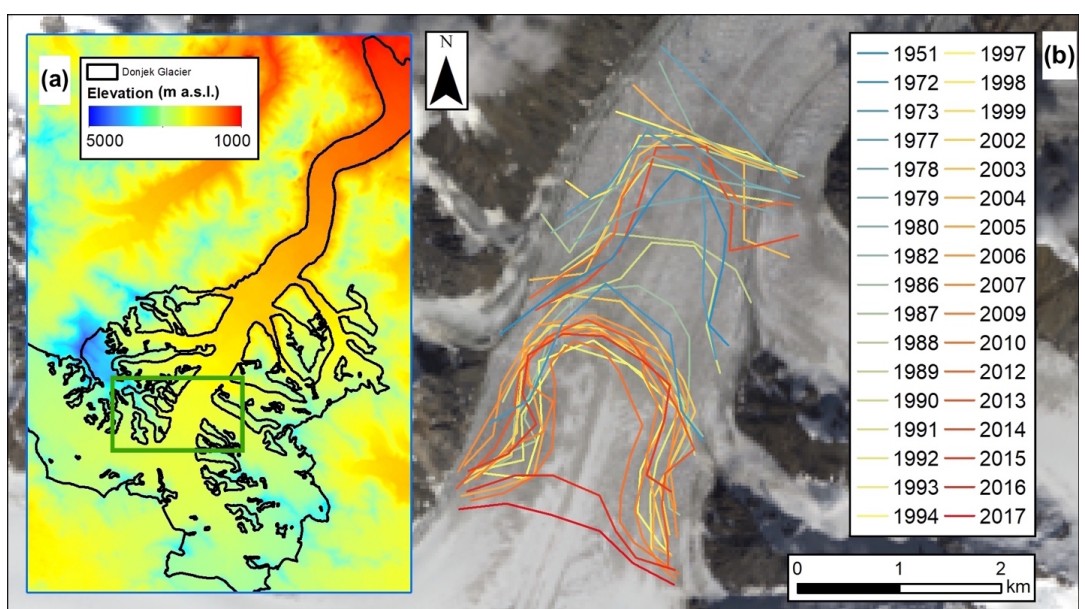


Figure 6. Donjek Glacier snowline. (a) Green box indicates extent of b, black outline shows
extent of Donjek Glacier on top of SPOT-5 DEM from 13 September, 2007. (b) Snowline from
1951 (blue) to 2017 (red). Satellite image from Landsat 8, 15 August, 2017.





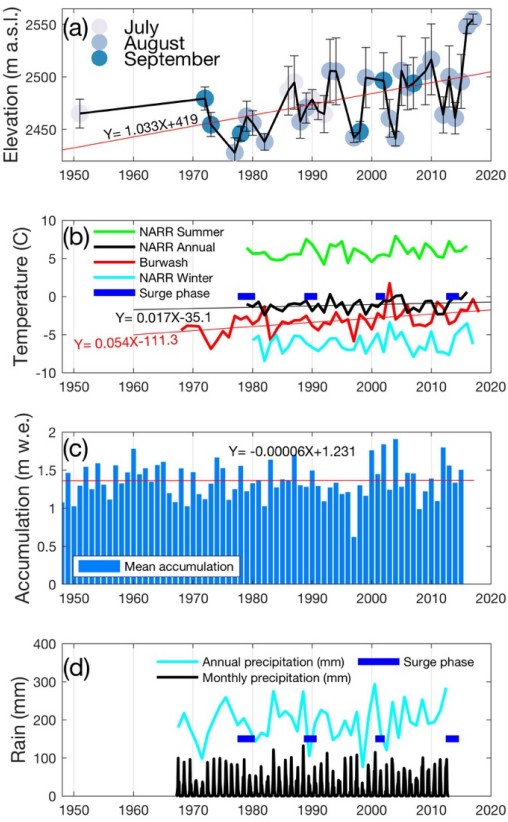


Figure 7. Donjek Glacier climate. (a) Snowline measurements from the last available satellite
image of each year in July (light blue), August (medium blue), and September (dark blue) with
black error bars indicating one standard deviation. Red line shows linear trend for study period.
(b) Burwash Landing annual average temperature record (red) with linear trend (thin red) and
NARR temperatures from 1400-1600 m on Donjek Glacier from winter (light blue), summer
(green), and annual mean (black) with linear trend (thin black). (c) Mean accumulation record
from 1995, 2002, and 2016 ice cores from Eclipse Icefield (blue bars) with linear trend (red). (d)
Rain from Burwash landing with annual (cyan) and monthly (black) totals from 1967 to 2012.
Blue bars indicate a period when Donjek Glacier was known to surge, time periods found by
Kochtitzky et al. (In Review).



### 4.4 Glacier geometry

Based on ground penetrating radar depth measurements downstream of the dynamic balance line
(8 km from glacier front; Fig. 1), we measured a bedrock rise towards the terminus (Figure 8).
Here, bedrock elevation rises from 810 to 890 m a.s.l. over a distance of 700 m in the
downstream direction, causing a 6.5° reverse bedrock slope (Figure 8), although the full spatial
extent of this reverse slope is unclear due to lack of measurements further down-glacier. The ice
thickness in this region ranges from 360 to 470 m, with deeper ice located closer to the dynamic
balance line.

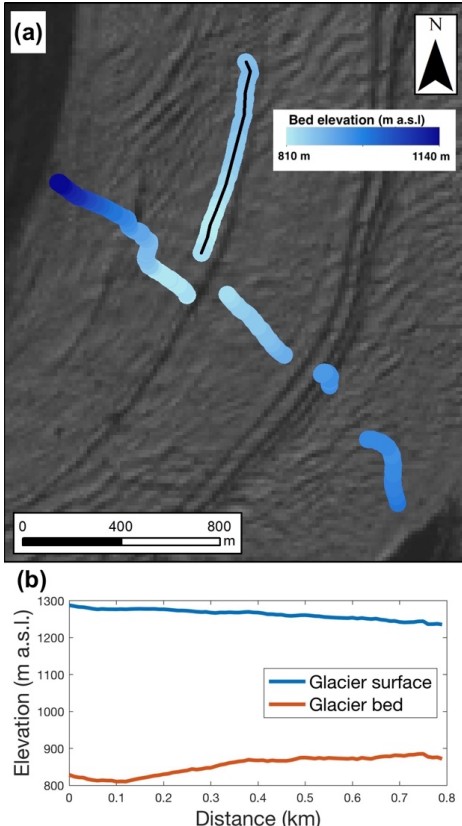


Figure 8. (a) Donjek Glacier bed elevations in the reservoir zone, which indicate a reverse slope
towards the terminus. Base image: Landsat 8, 23 September, 2017. Extent of figure indicated by
green box in Figure 1a. (b) Profile for along-flow GPR transect of surface ice (blue) and bedrock
(red). Extent of profile indicated by black line in 7a.



### 4.5 Temperature and precipitation patterns

An increase in mean annual temperature of ~0.05°C yr$^{-1}$ occurred at Burwash Landing between 1968 and 2017, equivalent to a total rise of ~2.5°C over the 50-year study period (Figure 7b), which is consistent with the measured rise in snowline elevation. The mean annual temperature at Burwash Landing reached a minimum of -6.9°C in 1973 and a maximum of 1.7°C in 2003. In addition to mean annual temperature rise, the number of cumulative positive degree days at Burwash Landing increased during each of the past four quiescent phases, from 15,095 PDD in the 1967-1977 quiescent period to 18,899 PDD in the 2001-2012 period (Figure 4a).

Because Burwash Landing can frequently be impacted by winter time temperature inversions and the temperature record has some missing data, we additionally examined the temperature record from NARR for the 1400 to 1600 m elevation range of Donjek Glacier, which corresponds to the area between ~8 - 13 km up glacier from the terminus. We find a smaller rise in temperature of ~0.02°C yr$^{-1}$ with an overall temperature rise of ~0.65°C over the period 1979 - 2016. Given a rise in the snowline elevation of ~70 m, this suggests the NARR temperature rise results are more accurate for Donjek Glacier compared to the temperature rise observed at Burwash Landing.

Annual snow accumulation derived from the Eclipse Icefield ice cores shows no significant trends over the study period, with values ranging from 0.62 to 1.91 m w.e. yr$^{-1}$ (Fig. 7c). A linear fit to the annual average accumulation from 1948 to present has a non-significant positive slope of 0.6 mm w.e. yr$^{-1}$ (95% confidence). However, the accumulation variance has increased from 0.039m$^2$ (1948 to 1982) to 0.068 m$^2$ (1982 to 2016) in recent decades.

Precipitation records from Burwash Landing indicate that the initiation of the 1988, 2000, and 2012 surges have coincided with several of the rainiest years on record, although not necessarily high accumulation years (Figure 7d). The top five annual rainfall totals on record from 1967 to 2012 for Burwash Landing were 2000 (293.5 mm), 2012 (284.0 mm), 1983 (274.9 mm), 1988 (273.9 mm), and 2005 (260.0 mm). However, the 1977 surge initiation coincided with relatively dry conditions (27th highest annual total rainfall year in the study period) (Figure 7d). Days and/or months with higher total rainfall could have occurred while the station at Burwash Landing was not operation.

Three of the top ten rainiest months appear to coincide with surge onsets (Figure 7d). The rainiest month on record was July 1988 (131.8 mm) and Donjek started surging the following





month (Kochtitzky et al., In Review). The third rainiest month on record occurred in August
2000 (114.7 mm) and Donjek started to surge that month or the next (Kochtitzky et al., In
Review). Donjek surged at the end of the 1960s (Kochtitzky et al., In Review) and the tenth
rainiest month on record occurred in July 1967.
**5.       Discussion**
**5.1 Snow and mass accumulation on surge-type glaciers**
The time it takes for a glacier to build up to its pre-surge geometry depends on the initial ice
volume displacement in the reservoir zone, the subsequent reservoir zone cumulative mass
balance, and the flux imbalance between actual and balance flux during quiescence (c.f. Clarke,
1987). Eisen et al. (2001) found that Variegated Glacier's cumulative mass balance consistently
reached a threshold of 43.5 m ice equivalent (39.9 m w.e.) before the glacier surged, while
Dyugerov et al. (1985) similarly found that a total of 360 ±70 million tons of mass accumulated
between each of four surge events of Medvezhiy Glacier, Tajikistan. On Donjek Glacier, 15.5 ±
1.46 m w.e. or 16.6 ± 2.0 m w.e. accumulates at Eclipse Icefield between surge events,
dependent on whether we account for an offset in redistribution to the surge initiation region,
~32 km downstream of the ice core site in Eclipse Icefield. For some glaciers, however, it is
known that during surges not all mass accumulated in the reservoir zone is emptied during a
subsequent surge: in Dyngjujökull, Iceland, for example, 13 km$^3$ of the 20 km$^3$ of ice
accumulated in the reservoir zone during its 20-year quiescence was transported to the receiving
zone in the 2 years of active surging (Björnsson et al., 2003). In addition, it is possible that the
consistent net accumulation observed at Eclipse Icefield between surge events simply reflects
consistent average accumulation (Wake et al., 2002; Kelsey et al., 2012) over each of the ~12-
year surge intervals (Abe et al., 2015; Kochtitzky et al., In Review).

Surges of glacier systems with surge-type tributaries, or mass advection to or from

adjacent basins (e.g., outlets from ice caps), can be irregular, and it can be difficult to relate a
surge interval to climatic conditions and accumulation rates, even under quasi-stable climatic
conditions (Glazovskiy, 1996; Björnsson et al., 2003). One of Donjek Glacier's tributaries surged
in 2004 and 2010 (~23 km from its terminus on the east side of Donjek's main trunk, shown in
Fig. 1), adding mass to Donjek's main trunk ~2 km upstream of the top of the trunk's reservoir
zone (Kochtitzky et al., in review). However, even though both these tributary surges occurred in

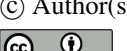



the quiescent phase of the trunk glacier, the tributary surges do not seem to have shortened the
duration of the trunk's quiescent phases.

**5.2 Climate and surge behavior**
Surge-type glaciers occur preferentially, but not exclusively, in specific climate zones that are
bounded by temperature and precipitation thresholds (Sevestre and Benn, 2015). Temporal
changes in surge controls, and thus in surge propensity, can occur due to climate change or
climate-forced changes in glacier size, elevation, hypsometry, thermal regime and/or subglacial
drainage system. Glaciers have been observed to change their surge behavior to being less
vigorous or complete cessation in some regions (Hoinkes, 1969; Frappé and Clarke, 2007;
Hansen, 2003; Christoffersen et al., 2005; Clarke, 2014), while widespread renewed surge
activity has recently occurred in the high Karakoram (Hewitt, 2007; Copland et al., 2011;
Quincey et al., 2011; Bhambri et al., 2017). This suggests that mass balance, melt conditions,
thermal regime and related supra-, en- and subglacial hydrology may all influence surging (e.g.
Dowdeswell et al., 1995; Eisen et al., 2005; Sund et al., 2009).

Although temperature is increasing by 0.05°C per year at Burwash Landing, and Donjek

Glacier has a negative mass balance, we do not observe an altered surge recurrence interval. Ice
cores from Eclipse Icefield also show no significant change in precipitation since at least 1950
(Wake et al., 2002; Kelsey et al., 2012). However, Kochtitzky et al. (In Review) report that each
of the past 8 surges have been aerially less extensive than previous surge events, similar to other
glaciers in the St. Elias, such as Lowell Glacier (Bevington and Copland, 2014). Less extensive
surge events are likely caused by a rising snowline and increasing number of positive degree
days, leading to a persistently more negative mass balance (Berthier et al., 2010; Larsen et al.,
2015). Similar observations of less extensive surge events during a period of negative mass
balance have occurred in Iceland (Sigurdsson and Jónsson, 1995). These observations suggest
that glacier wide mass balance controls the intensity of each surge event, while other
mechanisms control the surge recurrence interval.

Rapid mass redistribution, related surface lowering, and frontal advance, during surges

are important for short- and long-term glacier surface mass balance. Post-surge accelerated
ablation, thinning and retreat rates have been measured and modeled for surge-type glaciers in
Iceland (Adalgeirsdottir et al., 2005), West Greenland (Yde and Knudsen, 2007), Alaska



(Muskett et al., 2008), and Svalbard (Nuttall et al., 1997; Moholdt et al., 2010). For Donjek
Glacier, surges lead to glacier-wide negative mass balance (Kochtitzky et al., In Review). While
many of these glaciers are already experiencing a negative mass balance (Larsen et al., 2015), the
enhanced negative mass balance associated with surging should be taken into account in
projections of glacier mass loss in a changing climate.

**5.3 Surge onset and weather**
Weather has been suggested to affect surge initiation and termination (Harrison and Post, 2003);
in particular, strong melt, heavy rainfall, and large annual accumulation rates. Here, we focus on
surge initiation, as our results show that three of the top ten rainiest months at Burwash Station
coincided with surge onsets of Donjek Glacier.
Lingle and Fatland (2003) postulated that a temperate glacier will not surge until it has
built-up critical thickness (basal shear stress), and surface meteorological conditions occur that
store a large volume of water englacially. For Alaskan-type surges this has been shown to result
in a late-winter to spring surge onset (Raymond, 1987; Harrison and Post, 2003). A suite of
anecdotal evidence supports this hypothesis (Kamb et al., 1985; Muskett et al., 2008; Pritchard et
al., 2005), but there are also examples of temperate glaciers with surge initiation in seasons other
than winter (Harrison et al., 1994; Björnsson et al., 2003), including Donjek Glacier. Because
Donjek Glacier surge initiation always initiates during summer months (Kochtitzky et al., 2019),
it appears that seasonality is important to initiative a Donjek surge event. Rainfall may play an
important role in this process.
Surge initiation in polythermal glaciers may not be as dependent on the influx of surface
meltwater, but rather on reaching a critical thickness combined with water trapped at the bed.
Although a spring start is also common for polythermal glaciers (Hodgkins, 1997; Jiskoot &
Juhlin, 2009), these surges can potentially start in any season and may therefore still involve
enhanced snow or rainfall (Quincey et al., 2011). Surge trigger zones in polythermal glaciers
have also been correlated with ponding of water and extensive slush flows associated with heavy
late-spring (wet) snowfalls alternated with short-term episodes of exceptionally high
temperatures (Hewitt, 2007).  In summary, although some evidence and intuitive reasoning
suggest that the seasonality of surges could indeed be different for temperate glaciers than for



polythermal glaciers, no comprehensive analysis of seasonality of surge initiation and
termination in combination with thermal regime and surge development exists to date.

**5.4 Donjek surge mechanisms**

Abe et al. (2015) suggested that the constriction at 21 km from the terminus plays a

crucial role in causing Donjek Glacier to surge. However, Kochtitzky et al. (In Review) show
that the constriction was rather the upper extent of surge-type behavior, and in addition was
coincident with a change in bedrock lithology. We find no single conclusive factor that causes
Donjek Glacier to surge, although we can conclude that positive degree days are not a significant
control on surge recurrence interval. While Donjek Glacier reaches a consistent 13.1 to 17.1 m
w.e. accumulation before a surge event, this number cannot be confidently linked with the surge
recurrence interval given that it could also be an indicator of consistent decadal averaged
accumulation. Even though we show refilling of the reservoir zone on Donjek Glacier, limited
elevation measurements during recent surge events are inconclusive to use the reservoir zone as a
predictor for future surge events without more data. Assuming that past accumulation is an
indicator of future surge events, as displayed in Figure 4b, then the next surge is likely to occur
between 2022 and 2026.

More observations of Donjek Glacier surge kinematics, bedrock, and valley geometry are

needed to understand surging kinematics. While we show a bedrock rise beneath the dynamic
balance line, the relationship between the rise and surging is presently unclear. For a glacier in
the nearby Donjek Range, Flowers et al. (2011) suggest that its bedrock rise facilitates surging
because the reverse slope resists ice flow and promotes mass accumulation in the surge reservoir
zone during quiescence. Björnsson et al. (2003) conversely suggest from modeling results that
over-deepenings and reverse bed slopes enhance hydraulically inefficient subglacial drainage on
two surge-type glaciers in Iceland, diminishing mass accumulation. The role of the bedrock rise
in the surge behavior of Donjek Glacier is presently unknown, although it almost certainly plays
a role in controlling near-terminus ice dynamics, and thus is likely also involved in surge
dynamics.

**6.        Summary and Conclusions**





We use three ice cores to reconstruct the accumulation record for Donjek Glacier leading up to
seven documented surge events since the 1930s. We find that Eclipse Icefield received between
13.1 and 17.7 m w.e. (mean of 15.5 ± 1.46 m w.e.) total accumulation between surge events.
While mean annual air temperatures increased by 2.5°C from 1968 to 2017 at Burwash Landing,
30 km from Donjek Glacier terminus, we observe no change in the surge recurrence interval over
this time period, although each recent surge advance has become less extensive than the
previous. Although we find that cumulative accumulation is the most consistent climate variable
between surge events of Donjek Glacier, our results remain inconclusive as to the role of
accumulation in driving surge behavior. We suggest that yet unknown subglacial processes,
possibly including changes in till deformation rates, are the primary driver of surging at Donjek
Glacier, but mass accumulation remains a necessary precondition for a surge to initiate.

Satellite glacier surface elevation measurements reveal rapid refilling of the surge

reservoir zone 8-21 km from the terminus of Donjek Glacier within the first 2 years following a
surge event. We find almost no reservoir zone refilling occurs in the 5 years leading up to a surge
event. The reservoir zone thickening is not the only cause of surge initiation, and therefore a
critical basal shear stress may need to be coincident with a hydrological switch. The highest
rainfall amounts typically occur during the summer month preceding a surge initiation. While not
every observed surge initiates with a high rainfall amount, the three most recent surges (1988-
1990, 2000-2002, 2012-2014) all coincide with one of the top five years on record for
precipitation.

Even though we observe a bedrock rise in the receiving zone of Donjek Glacier,

downstream of the dynamic balance line, the role that overdeepening and a reverse bedrock slope
play in surging of Donjek Glacier remains a crucial question. Further observations of bedrock
and bed elevation are necessary to understand surge mechanisms of Donjek Glacier. Monitoring
surface elevation changes on Donjek Glacier as it prepares for its next surge event by the mid-
2020s can yield valuable knowledge about how the subglacial hydrology beneath Donjek Glacier
changes as a surge initiates. This will ultimately lead to more knowledge of surge initiation
mechanisms, which can lead to better forecasting of surge events and magnitudes and therefore
mitigate glacier hazards.

**Author Contribution**



WK, KK, LC, and HJ designed the study. WK carried out data analysis for all remote sensing
and weather station data. DW, EM, KK, WK, and SC collected and analyzed the 2016 ice core.
DW, EM, KK, and WK reprocessed the 2002 ice core. LC, WK, BM, and CD collected glacier
thickness measurements. SW downscaled the NARR dataset and did all associated data analysis.
WK prepared the manuscript with contributions from all co-authors.

**Acknowledgements**

WK is supported by the National Science Foundation Graduate Research Fellowship under Grant
No. DGE-1144205. WK thanks Dan and Betty Churchill for funding the 2017 field season and
Geophysical Survey Systems, Inc. for funding the 2018 field season. LC thanks the Natural
Sciences and Engineering Research Council of Canada, University of Ottawa and Polar
Continental Shelf Program for funding and field logistics. KK thanks the NSF for funding St.
Elias research, NSF AGS-1502783. CD thanks the Canada Research Chairs Program, Natural
Sciences and Engineering Research Council of Canada, and Polar Continental Shelf Program for
funding and field logistics. WorldView DEMs provided by the Polar Geospatial Center under
NSF-OPP awards 1043681, 1559691, and 1542736. We thank Daniel Dixon, Steven Bernsen,
Justin Leavitt (UMaine), Dorota Medrzycka (uOttawa), and Patrick Saylor (Dartmouth College)
for their help collecting the 2016 ice core, Douglas Introne and Michael Handley for ice core
analysis, and Cameron Wake (University of New Hampshire) for help reprocessing the 2002
core isotope data. We thank Robert McNabb (uOslo) for the 2002 ASTER DEM and Icefield
Discovery, Trans North helicopters and the Kluane Lake Research Station for their logistical
support.

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
