# Peer review of "The Impact of Climate on Surging at Donjek Glacier, Yukon, Canada"

_The Cryosphere, 2019_

## Referee Comment (RC1) · Anonymous Referee #1 · 28 May 2019

General comments

Glacier surges have been shown to cluster non-randomly in many mountain regions of the world but the underlying processes remain little understood. This manuscript presents an innovative approach to the study of the climatic forcing of glacier surges in Donjek glacier, Canada, with a combination of ice core, meteorological, remote sensing and GPR data, giving an strong insight on the glacier surging main forcings. The work compiles existing data from a series of previous works of this well studied glacier, and builds on with significant new evidence. The results are convincing and conclusions seem particularly strong because they apply to seven surge cycles. Discussion is valuable and focusses on the interpretation and some limitations of the results.

I have some comments regarding a small part of the methods, some of the writing and

some of the graphs, which are detailed in the following sections.

For these reasons I suggest the paper should be accepted with minor revisions.

Specific comments

In the methods section it is clear that the multimethod approach relies on other work which is duly cited. Nevertheless I consider it is important to briefly and clearly outline the methods used in the datasets analyzed in this manuscript, so that the reader does not need to constantly refer to other papers. In particular organization of the ice core and remote sensing methods and materials could be clarified and improved.

One methodological observation I formulate is the interpretation of a small section of the snowline as representative of the entire glacier. I propose a semiautomatic method that could provide more robust glacier-wide snowline data.

Another methodological observation is the interpretation of the extent of reservoir data. I think it is important to show some complete longitudinal profile of the glacier elevation change, so that the area where the flowing instability occurs is clearly identified (e.g.: Burgess and others, 2012, Fig. 3; Pitte and others, 2016, Fig. 4). I could not help to notice strong elevation changes above the proposed threshold of 20 km (Fig 5) and wonder how much higher it originates. Related to this point, once the reservoir and receiving area are clearly established I suggest calculating the displaced volumes based on the areal data of these two zones. A match of these volumes, within error bars, would strengthen the interpretation of the abovementioned zones.

The graphs are generally hard to read, consider widening the figures and using larger font. Also the graphs and maps have a reference box, but much of the information is repeated in the caption. I suggest taking advantage of the reference and keeping the captions shorter.

Another suggestion is that, for clarity, multiple bar plots be replaced by lines, which might facilitate the comparison between the different surge cycles or variables represented.

Technical corrections

Line 1. Consider the use of capitals only for proper names.

Line 99. Figure 1a. Include elevation of some summit, Eclipse icefield met station and/or glacier front to give and idea of the elevation range. Figure 1b. In the location map, considering including a shading for mountain areas (e.g. over 2000-3000m and/or glacier distribution). It is not necessary to repeat in the figure caption the information already in the reference (Eclipse station, 5 km marker, etc.).

Line 107. Include some more information regarding glacier classification, shape and morphometry (elevation range). Also here or in the final part of the introduction include the number/area of glaciers in the region and the number of surging ones.

Line 114. Please remove this last sentence. Ice coring is extensively described in methods. Consider including a paragraph descrining climatic setting.

Line 117. Please rephrase this paragraph to describe events in chronological order.

Line 129. In Figure 2 the ordinate axis (y) is not really used. Consider including a variable such as glacier area, length of velocity to give an idea of the different surge magnitudes (e.g.: Donjek et al 2016, Fig 2a).

Line 135. This paragraph is crowded with data which makes the reading hard. Separate in two paragraphs: one for general description of the cores and another for describing dating methods. Consider including a table with the main metadata of the ice cores (date of collection, length, age range, dating methods and reference).

Line 155. Insert: "equation 1"

Line 156. Start by mentioning the aim of this step: "In order to obtain an annually-dated timescle, five individuals. . .".
Line 176. Figure 3, considering using full page width for this figure. The two panels could be combined (e.g.: Ginot et al 2006, fig 3). To facilitate inter-comparison, considering plotting the original series shaded in the background and moving averages in the foreground.

Line 203. Again this paragraph is crowded with data. Add horizontal resolution and source to Table 1 and focus on the comparative aspects of the different elevation datasets.

Line 223. The method used to identify the glacier ELA is rather sui generis. The snowline separates the entire accumulation zone from the entire ablation zone. This is a very large glacier and although the snowline of the trunk glacier (as used in Fig 6) might be representative of the whole glacier, this is not proven. I suggest using, for the selected images, a standard method (thresholded band ratio or NDSI) to obtain the snow covered area of the entire glacier and use the average lower boundary (i.e.: Kargel et al 2014 section 4.3.3.5 and 4.3.3.9, Rabatel and others, 2012).

Line 295. Figure 4. Consider using lines for multiple bar plots. Bar plots are adequate to represent accumulation but when one or two detailed series are plotted together comparison is tricky. In particular, to identify lines intersections were the different records inter-consistency changes. Consider using the yearly average (black continuous and dashed lines). This would allow analyzing differences in accumulation rates during the different buildup phases. Do not repeat in the caption information given in the graph reference.

Line 321. The interpretetaion of the reservoir are could be supported by additional evidence if the volume loss of the reservoir area and the volume gain of the receiving are were calculated and found to be similar within error bars. I suggest including this calculation in this section.

Line 322-326. This lines should be moved to methods (were part of this is already explained).

Line 343. The snowline for the entire glacier should be calculated and discussed, especially if the AAR is to be considered. See comment in line 223.

Line 346. In climatology it is customary to present decadal rates. Check manuscript for consistency.

Line 354. This figure is a little too busy. Fig 6b some elevation data would be welcome to illustrate the elevation change range of the ELA. Consider including the colomap, hillshaded relief in this figure. The full set of snowlines is of little use other than showing the detailed work (maybe put it in SM). Instead, consider showing a limited number full glacier AAR in a set of panels to show variability. The full time series is already shown in Fig 7a. Figure 6a, topographic information of the entire glacier should be included in an enlarged version of Fig 1 as contours and point elevation values rather than as an insert here.

Line 361. Fig 7 consider including the fit ($r^2$) of the linear regressions. Inter-annual precipitation variability is usually not adequately represented by linear trends. Linear trends are also highly sensitive to first and last year of the records, this effect can be quite strong in discontinuous datasets such as the snowline (Fig. 7 a). It short it might not be the most adequate parameter to find a trend in such variables. Fig 7b. Temperature should be expressed as anomalies so the bias of the different records is removed and they can be more easily compared. Figure 7c and d, consider including a running average with a 10-12 window that could highlight the variability at the surge-cycle scale.

Line 378. Figure 8. Provide a caption of the location of the transects here rather than in Fig 1. It is great to see some direct field measurements of glacier thickness and bed topography, yet the relevance depends on the accurate definition of the reservoir area (see Specific comments).

Line 382. Consider the tittle "Temperature and precipitation trends", patterns would be more adequate for analysys of the geographic distribution of the variables

Line 395. This is a bit simplistic since ELA depends on both temperature and precipitation.

Line 402. Verify if the unit is m.e. instead of m$^2$.

Line 405. After enlarging consider including 5 yr markes on the x axis of Fig 7.

---

## Short Comment (SC1) · 7 Jun 2019

This comment is focused primarily on section 4.3. I commend the authors for using a combination of ice core accumulation records, transient snowline observations and ice surface elevation change to examine the relationship between mass balance, climate variables and surge cycle. These are independent observations of surface mass balance and consequent transmission downglacier. The snowline section can provide more context and a stronger record that ties into the declining mass balance and reduced surge volume.

344: Suggested clarification and supporting references. . . . Transient snowline should be used instead of simply snowline. "Our remote sensing analysis illustrates that the

summer transient snowline (TSL) in the center flow unit of Donjek Glacier has migrated up-glacier by 55 m yr-1 horizontally and risen by ∼1.0 m yr-1 in elevation over the period 1951 to 2017 (Figures 6 and 7a). The increase in TSL elevation during this period has been observed on other glaciers in the region including Brady Glacier and Taku Glacier (Pelto et al, 2013a and 2013b). Over the study period the TSL was lowest in 1977 (Figure 7a), with an accumulation area of 337.3 km2 and an Accumulation Area Ratio (AAR) of 75.3%. The TSL reached its highest average elevation of ∼2550 m a.s.l. in 2017, corresponding to an AAR of 68.4%. The higher TSL is indicative of a reduced surface mass balance."

349 The following statement needs supporting details to illustrate how Donjek Glacier is different based on the observation dates when the maximum TSL is achieved and if there is any trend in the timing of the TSL or any migration rate observations available, that would support the statement below.

"Even though some TSL measurements were made early in the ablation season, we do not find our TSL measurements to be biased by timing of the observation, as TSL elevations in the late melt season were not consistently different from those early in the melt season (Figure 7a)."

Suggested considerations: Define early melt season versus late melt season. Has there been a shift in the timing of TSL maximum? Is there a migration rate that can be identified for July and for August?

The TSL does migrate upward through the summer on many Alaskan glaciers. The migration rate has been observed Brady and Taku Glacier for the July-September period. On Taku Glacier "Mean rise of the TSL for 16 periods averages 3.7 md−1 during the July–September period, for the elevation range between 750–1100 m" (Pelto et al 2013a). On Brady Glacier the TSL migration rate is 3.6 md-1, the snowline has also risen 145 m (Pelto et al 2013b). The elevation range observed for this migration has been from 900-1300 m. The elevations are much higher on Donjek Glacier which also

has a less temperate climate. The result is the maximum TSL elevation Is achieved earlier in the year, this does need to be better established. The earlier maximum is evident in 2015 the TSL reaches a maximum on July 25 and is lower by Aug. 3 after a summer snow event. In 2017 the TSL is above 2500 m by late July and shifts little up to Aug. 15 as noted by the authors, but is lower by 8/24 after a late summer snow event. In 2018 the snowline again reaches +2500 m by Aug. 1 and by Aug. 11 is again lowered by a late summer snow event. This implies there is a limited TSL migration rate after late July.

539: It is noted that the timing of surge events has been consistent, but the size has continued to decline. Should it be added that "This decline in surge volume coincides with a rising snowline indicative of reduced mass balance." The reduced mass balance is evident regionally Das et al (2014) and Larsen et al (2015) as well as on Donjek Glacier. This could be observed at 465 too.

Das, I., Hock, R., Berthier, E., and Lingle, C.: 21st-century increase in glacier mass loss in the Wrangell Mountains, Alaska, USA, from airborne laser altimetry and satellite stereo imagery. Journal of Glaciology, 60(220), 283-293, doi:10.3189/2014JoG13J119, 2014.

Larsen, C. F., Burgess, E., Arendt, A. A., O'Neel, S., Johnson, A. J., and Kienholz, C.: Surface melt dominates Alaska glacier mass balance. Geophys. Res. Lett., 42, 5902–5908, doi:10.1002/2015GL064349, 2015.

Pelto, M. , Capps, D. , Clague, J. and Pelto, B.: Rising ELA and expanding proglacial lakes indicate impending rapid retreat of Brady Glacier, Alaska. Hydrol. Process., 27: 3075-3082. doi:10.1002/hyp.9913, 2013.

Pelto, M., Kavanaugh, J., and McNeil, C.: Juneau Icefield Mass Balance Program 1946–2011, Earth Syst. Sci. Data, 5, 319-330, https://doi.org/10.5194/essd-5-319-2013, 2013.

---

## Referee Comment (RC2) · Anonymous Referee #2 · 2 Jul 2019

This paper addresses an interesting glaciological problem and presents a wealth of data on Donjek Glacier and its climatic setting. However, it is frustratingly inconclusive, and sheds little light on surging processes at the site. This reflects a systemic problem with the paper, particularly a disconnection between the research aims and what the data sets are capable of showing. The data do point the way to some interesting possibilities, but these require further investigation before their potential can be fully realized.

A major issue is the question of whether the timing of surge onset is related to cumulative snow accumulation, as has been demonstrated at Variegated Glacier by Eisen et al. (2001). A large proportion of the paper (pp. 133-200, 287-319, 417-444) is devoted to this question, using firn- and ice-core data to construct time series of annual

accumulation and comparing them to the intervals between surge events. However, as the authors point out, at Donjek Glacier the surge reservoir zone is in the ablation area, ~15 km downstream of the snowline. Thus elevation changes are driven not by snow accumulation, but by the flux divergence minus surface ablation rates. It is clear that the flux divergence term is dominant in this case. During surge build-up, the ice thickens due to convergent flow (which more than offsets melt), while during the surge phase the ice thins faster than the melt rate due to divergent flow. There is no reason to expect that dynamic thickness changes in the ablation zone during surge build-up are correlated with annual accumulation rates some 30 km upstream, even when the latter are adjusted to account for transport time. Ice is not delivered to the reservoir zone one annual increment of snowfall at a time. The integrated mass balance upglacier of km21 will determine the balance flux, but the distances involved mean that the balance flux will be insensitive to annual variations in accumulation.

Unfortunately, this means that all the careful work with cores to determine annual variations in snow accumulation is to no avail. The correlation between cumulative accumulation and surge period has no direct physical meaning, and Figure 4 does not show the same thing as Eisen's data. I think the authors get it right when they concede that "it is possible that the consistent net accumulation observed at Eclipse Icefield between surge events simply reflects consistent average accumulation over each of the ~12-year surge intervals" (432-435).

A more fruitful line of inquiry would be to examine time-series of elevation changes in the reservoir zone itself, and to relate these to dynamic cycles. At present, there is no discussion of the important observations by Abe et al. (2016), which revealed consistent velocity patterns over the last two surge cycles. What are the causal relationships between ice dynamics and elevation changes, and how do they evolve over the surge cycle? How & when do the dynamics influence elevation change (e.g. through cycles of divergent vs. convergent flow), and how & when do elevation changes influence the dynamics (e.g. via changing shear stresses or other factors)? The ice surface elevation

change data presented in the paper are obviously too sparse to examine these issues in detail, but a good starting point would be to look at the trends in ice elevation shown in Fig 5 alongside the dynamic patterns revealed in Abe et al's velocity records (their Fig 1c and d). In particular, note how the patterns of elevation change correspond to the spatial patterns of velocity during quiescence (and, importantly, the variations in glacier width, particularly the 1/3 reduction in glacier width between km22 and km18). The narrowing of the glacier indicates that the glacier needs to speed up below km20 to satisfy continuity, but it either undershoots (quiescence) or overshoots (surge) the required value. Why should this be?

In a throwaway statement on line 510, we read that the upper extent of surge behavior is "coincident with a change in bedrock lithology". The fact that the dynamic instability occurs at both a topographic and geologic transition deserves to be investigated and reported in detail. What exactly is the lithological change? How does this relate to valley morphology? How might this affect the ability of the glacier to evacuate basal meltwater? If they hope to understand the surging behavior of Donjek Glacier, the authors need to give serious consideration to the idea that the instability relates to topography and/or geology and how it interacts with the dynamics. Knowledge of the subglacial topography would add a great deal of value in this respect, but the current sampling shown in Fig. 8 is far too sparse to allow any useful analysis.

Of course, the topic of the paper is the impact of climate on surging, and the fact that recent climate changes have had no discernable impact on the length of the surge cycle is clearly of interest. But the significance of this can really only be understood when considered alongside the causes of the dynamic instability, and the question of why the instability should be resilient to the observed climate change.

The subject of surge onset and weather shows promise, but this too will require more work. The authors state "three of the top ten rainiest months appear to coincide with surge onsets" (410). This of course means that the other seven rainiest months do not, but there could be something in this and it should be investigated in more detail.

Perhaps rain events do have an influence, but only if the glacier is 'primed', or close to a critical state (i.e. if the surge front is close to the terminus, or some other condition is met). Can a particularly rainy month trigger a surge a year or two sooner than average? Conversely, can a dry summer delay surge onset causing the period to be longer than average? Detailed figures showing monthly precipitation totals relative to surge onsets would help shed light on this (resolution of the monthly precipitation record in Fig 7d is currently too low to convey useful information).

At present, the paper's shortcomings and 'red herrings' mean that it does not advance our understanding of the timing of surges with regard to environmental controls, or yield new insights into surge mechanisms at the site. This is a pity, because clearly a great deal of work has gone into the data collection, analysis and presentation. There is potential for something of value here, especially with regard to the possible topographic and geological controls on the instability in the ablation zone of Donjek Glacier, and the timing of surge onset relative to enhanced inputs of surface water. My recommendation is that the authors set the paper aside until they have explored these issues, and got closer to the heart of the intriguing problem of Donjek's unusual surging behavior.

---

## Author Comment (AC1) · 29 Aug 2019

We appreciate the thoughtful comments from the two anonymous referees and one short comment from Mauri Pelto. Please see below for our response to all comments.

Comment from Referee: In the methods section it is clear that the multimethod approach relies on other work which is duly cited. Nevertheless I consider it is important to briefly and clearly outline the methods used in the datasets analyzed in this manuscript, so that the reader does not need to constantly refer to other papers. In particular organization of the ice core and remote sensing methods and materials could be clarified and improved. Author response: We will more explicitly state how the 1996 and 2002 cores were collected. We are unclear how to clarify the remote sensing methods fur-

[Figure]

ther. We cite data sources for DEMs and would rather not discuss how they were made, as this unnecessarily lengthens the paper. The important point is that the DEMs are available and we cite the sources. Changes in the manuscript: We will further discuss collections of 1996 and 2002 ice cores from Yalcin and Wake (2001) and Yalcin et al. (2007). We will change lines 140-141 to the following: The accumulation record from the 1996 ice core was originally reported by Yalcin and Wake (2001) and we use their original data here. They took a 160 m core and shipped it frozen to the University of New Hampshire where they used straight core processing and continuously sampled the core in 10 cm segments (Yalcin and Wake, 2001). An original depth-age scale for the 2002 core was developed by Yalcin et al. (2007). They extracted two cores, which we only use the longer, 350 m core, and they continuously sampled the cores in 10-15 cm segments for stable isotopes and major ions. We will incorporate additional clarifying language on picking the annual snowline based on suggestions from the short comment from Mauri Pelto.

Comment from Referee: One methodological observation I formulate is the interpretation of a small section of the snowline as representative of the entire glacier. I propose a semiautomatic method that could provide more robust glacier-wide snowline data. Author response: This is an excellent suggestion and we would like to see this as well. Unfortunately, this is extremely challenging for a glacier such as Donjek. We attempted a semi-automated method using band ratios, but the manual digitization ultimately proved more accurate. This is due to the large number of very small tributary glaciers and nunataks sticking out of the ice. We will show the snowline for the entire glacier that we digitized manually. Changes in the manuscript: We will update figure 6 to show the AAR and entire glacier snowline.

Comment from Referee: Another methodological observation is the interpretation of the extent of reservoir data. I think it is important to show some complete longitudinal profile of the glacier elevation change, so that the area where the flowing instability occurs is clearly identified (e.g.: Burgess and others, 2012, Fig. 3; Pitte and others, 2016,

Fig. 4). I could not help to notice strong elevation changes above the proposed threshold of 20 km (Fig 5) and wonder how much higher it originates. Related to this point, once the reservoir and receiving area are clearly established I suggest calculating the displaced volumes based on the areal data of these two zones. A match of these volumes, within error bars, would strengthen the interpretation of the abovementioned zones. Author response: We agree that these would be excellent ideas to demonstrate in this paper. However, we have recently described these changes in a newly published Journal of Glaciology article entitled "Terminus advance, kinematics and mass redistribution during eight surges of Donjek Glacier, St. Elias Range, Canada, 1935 to 2016" (https://doi.org/10.1017/jog.2019.34). This paper acts to fully define the reservoir and receiving areas and goes into more depth on these topics. The JoG paper describes some volume estimates for each zone, but it is difficult to get reliable measurements further upglacier due to the lack of distinguishing features to match in stereo satellite imagery over snow-covered areas. It is also unfortunately impossible to calculate total volume displaced as some of the crucial data are derived from LiDAR, which only provides centerline measurements. The reviewer is correct to point out the elevation changes above the surge-type portion of the glacier. We go into more detail in the JoG paper about this topic, but we believe this is the refilling of the reservoir zone. We speculate in that paper about this process, but lack observations to show it here. We will briefly summarize the elevation changes observed on Donjek, and described in our JoG paper. Changes in the manuscript: We will change the citation from Kochtitzky et al., In Review to Kochtitzky et al., 2019. We will extend the elevation profile displayed in figure 5 to include areas up to at least 40 km where the glacier splits into two main branches to show a more complete longitudinal profile. We will add a summary of the JoG paper elevation changes on line 445 to highlight the drop in elevation above 21 km from 2001-2007 and discuss the role of elevation changes (and mass movements) in the surging process.

Comment from Referee: The graphs are generally hard to read, consider widening the figures and using larger font. Also the graphs and maps have a reference box,

but much of the information is repeated in the caption. I suggest taking advantage of the reference and keeping the captions shorter. Author response: We will fix the figures throughout to improve the figure aspect and readability while still conforming to the journal's formatting requirements. Changes in the manuscript: We will increase the size of the font on figures, work on clarifying them where possible, and remove language in the figure captions that is repeated in the figure legends.

Comment from Referee: Another suggestion is that, for clarity, multiple bar plots be replaced by lines, which might facilitate the comparison between the different surge cycles or variables represented. Author response: If the referee is referring to figure 4, this idea was considered. We thought the bar plots were clearer given that we wanted to emphasize the cumulative snowfall. This also mimics previous work done on similar topics (e.g. Eisen et al., 2001). Changes in the manuscript: We will alter the size and text to increase the clarity of the figure.

Comment from Referee: Line 1. Consider the use of capitals only for proper names. Author response: We will change the title and check that proper names are capitalized elsewhere throughout the paper. Changes in the manuscript: Proposed new title "The impact of climate on surging at Donjek Glacier, Yukon, Canada"

Comment from Referee: Line 99. Figure 1a. Include elevation of some summit, Eclipse icefield met station and/or glacier front to give and idea of the elevation range. Figure 1b. In the location map, considering including a shading for mountain areas (e.g. over 2000-3000m and/or glacier distribution). It is not necessary to repeat in the figure caption the information already in the reference (Eclipse station, 5 km marker, etc.). Author response: We will include language about elevation along the glacier. Changes in the manuscript: We will add the following language around line 108: "The terminus of Donjek Glacier is located at ∼1000 m a.s.l. The highest point within the glacier is Mount Walsh at 4507 m a.s.l. Eclipse Icefield, where the cores were drilled, is at ∼3020 m a.s.l." We will remove repeated items in the caption and legend.

Comment from Referee: Line 107. Include some more information regarding glacier classification, shape and morphometry (elevation range). Also here or in the final part of the introduction include the number/area of glaciers in the region and the number of surging ones. Author response: See previous comment for elevation range additions. We will add language citing previous work done on surge-type glacier inventories in this region. Changes in the manuscript: We will include data and citations from Sevestre and Benn (2015) and Clarke et al. (1986) on the number of surge type glaciers in the St. Elias. We will also include more information about the Donjek Glacier elevation range.

Comment from Referee: Line 114. Please remove this last sentence. Ice coring is extensively described in methods. Consider including a paragraph describing climatic setting. Author response: We will remove the sentence on line 114. We will add information about the climatic setting. Changes in the manuscript: Delete line 114. We will add the annual average precipitation and air temperature for the weather station at Burwash Landing.

Comment from Referee: Line 117. Please rephrase this paragraph to describe events in chronological order. Author response: We will check the sentence for date order. Changes in the manuscript: We will check the wording to ensure everything is in chronological order.

Comment from Referee: Line 129. In Figure 2 the ordinate axis (y) is not really used. Consider including a variable such as glacier area, length of velocity to give an idea of the different surge magnitudes (e.g.: Donjek et al 2016, Fig 2a). Author response: We originally made this plot to mimic figure 2 from Eisen et al. (2001). The suggestion to add more data is a good one and would demonstrate the glacier change during a surge event, but unfortunately we currently don't have data to show the area, length, or velocity changes during this time period. We have some spotty data, which was published our recent JoG article, but do not have the full period change. Changes in the manuscript: We will include more references to Abe et al (2016) and our recent

JoG paper to talk about the changes observed over the course of the surge events and how the surges have changed through time.

Comment from Referee: Line 135. This paragraph is crowded with data which makes the reading hard. Separate in two paragraphs: one for general description of the cores and another for describing dating methods. Consider including a table with the main metadata of the ice cores (date of collection, length, age range, dating methods and reference). Author response: Thank you for this suggestion. Changes in the manuscript: We will split the paragraph at line 139 to separate the description and dating methods. We will also include a table as the referee suggests

Comment from Referee: Line 155. Insert: "equation 1" Author response: Thank you for this suggestion. To our understanding, the way we have formatted the equation is consistent with TC style: see 'Mathematical notation and terminology' under https://www.the-cryosphere.net/for_authors/manuscript_preparation.html Changes in the manuscript: None.

Comment from Referee: Line 156. Start by mentioning the aim of this step: "In order to obtain an annually-dated timescle, five individuals. . .". Author response: Thank you for the suggestion, we will add this language. Changes in the manuscript: We will add modify the start of the sentence on line 156: "To obtain an annually resolved time scale,. . ."

Comment from Referee: Line 176. Figure 3, considering using full page width for this figure. The two panels could be combined (e.g.: Ginot et al 2006, fig 3). To facilitate inter-comparison, considering plotting the original series shaded in the background and moving averages in the foreground. Author response: We will combine the two figures. Changes in the manuscript: We will plot figures 3a and b on the same plot and make sure that the data do not overlap too much to ensure that the figure is still readable and demonstrates the core results.

Comment from Referee: Line 203. Again this paragraph is crowded with data. Add

horizontal resolution and source to Table 1 and focus on the comparative aspects of the different elevation datasets. Author response: Thank you for this suggestion. The source is already indicated on table 1. This paragraph is intended to describe the datasets and how we acquired them, not compare them, but we will refine the wording to make it easier to follow Changes in the manuscript: We will add the spatial resolution of the datasets to table 1, and reword the text to ensure that it's easy to follow

Comment from Referee: Line 223. The method used to identify the glacier ELA is rather sui generis. The snowline separates the entire accumulation zone from the entire ablation zone. This is a very large glacier and although the snowline of the trunk glacier (as used in Fig 6) might be representative of the whole glacier, this is not proven. I suggest using, for the selected images, a standard method (thresholded band ratio or NDSI) to obtain the snow covered area of the entire glacier and use the average lower boundary (i.e.: Kargel et al 2014 section 4.3.3.5 and 4.3.3.9, Rabatel and others, 2012). Author response: We appreciate the comment here to ensure we are calculating a snowline for all of Donjek Glacier. We will update figure 6 to show the entire glacier snowline instead of just the snowline along the trunk. We will update the methods and results sections to describe the entire glacier snowline. Changes in the manuscript: We will change figure 6 to show the snowline for the entire glacier and describe these results in the methods section.

Comment from Referee: Line 295. Figure 4. Consider using lines for multiple bar plots. Bar plots are adequate to represent accumulation but when one or two detailed series are plotted together comparison is tricky. In particular, to identify lines intersections were the different records inter-consistency changes. Consider using the yearly average (black continuous and dashed lines). This would allow analyzing differences in accumulation rates during the different buildup phases. Do not repeat in the caption information given in the graph reference. Author response: Thank you again for this suggestion, but please see reply above to the same comment. We provide the annual accumulation in figure 7c, this shows how the accumulation varies from year to

year against the annual average. Changes in the manuscript: We will remove caption information that is in the figure legends.

Comment from Referee: Line 321. The interpretation of the reservoir are could be supported by additional evidence if the volume loss of the reservoir area and the volume gain of the receiving are were calculated and found to be similar within error bars. I suggest including this calculation in this section. Author response: Please see response above to previous comment and Kochtitzky et al. (2019) in Journal of Glaciology; unfortunately, we don't have the elevation data available to compute reservoir-wide volume changes Changes in the manuscript: We will include a more through discussion of the results from the JoG paper to describe the elevation changes.

Comment from Referee: Line 322-326. This lines should be moved to methods (were part of this is already explained). Author response: Thank you for pointing this out. Changes in the manuscript: We will remove lines 322-326 as they already appear in the introduction and are more appropriate there.

Comment from Referee: Line 343. The snowline for the entire glacier should be calculated and discussed, especially if the AAR is to be considered. See comment in line 223. Author response: We calculated the snowline for the entire glacier and derived the AAR from these measurements for 2017 and 1977. We will show these measurements in an updated figure 6, see more below. Changes in the manuscript: We will further discuss the snowline for the entire glacier in the methods section and update figure 6 to show the AAR as described below.

Comment from Referee: Line 346. In climatology it is customary to present decadal rates. Check manuscript for consistency. Author response: Thank you for pointing this out, we will show climate related data in decadal rates. Changes in the manuscript: We will change the numbers on lines 346, 383, and 392 to reflect decadal instead of annual rates.

Comment from Referee: Line 354. This figure is a little too busy. Fig 6b some elevation

data would be welcome to illustrate the elevation change range of the ELA. Consider including the colomap, hillshaded relief in this figure. The full set of snowlines is of little use other than showing the detailed work (maybe put it in SM). Instead, consider showing a limited number full glacier AAR in a set of panels to show variability. The full time series is already shown in Fig 7a. Figure 6a, topographic information of the entire glacier should be included in an enlarged version of Fig 1 as contours and point elevation values rather than as an insert here. Author response: Thank you for this suggestion, we will rearrange the figures to clarify the snowline and topography and better show the change in snowline through time. Changes in the manuscript: We will remove figure 6 as it stands now, we will replace it with a map of the AAR for 1977 and 2017 to show the change from the minimum to maximum year AAR. Figure 7a will remain the same to show the change in elevation through time. The AAR lines will be plotted on top of a topographic map to show the elevation range on Donjek Glacier and figure 1 will still display the introductory Landsat image.

Comment from Referee: Line 361. Fig 7 consider including the fit (r2) of the linear regressions. Inter-annual precipitation variability is usually not adequately represented by linear trends. Linear trends are also highly sensitive to first and last year of the records, this effect can be quite strong in discontinuous datasets such as the snowline (Fig. 7 a). It short it might not be the most adequate parameter to find a trend in such variables. Fig 7b. Temperature should be expressed as anomalies so the bias of the different records is removed and they can be more easily compared. Figure 7c and d, consider including a running average with a 10-12 window that could highlight the variability at the surge- cycle scale. Author response: For 7b we considered showing anomalies, but wanted to highlight the difference between the annual average temperature magnitude in Burwash Landing and on the glacier from the NARR dataset. Thus, we chose to show temperature magnitudes, not anomalies. For Fig. 7c we will include a 10-year moving average of annual accumulation. The point of Fig. 7d is to show the occurrence of high precipitation months coinciding with surge initiation, meaning that a moving average would detract from this objective. In addition, cumulative accumulation

seems to be most important for surge initiation, as already shown in Fig. 4. Changes in the manuscript: The r2 values of the best fit lines will be included in the figure caption. For Fig. 7c we will include a 10-year moving average of annual accumulation.

Comment from Referee: Line 378. Figure 8. Provide a caption of the location of the transects here rather than in Fig 1. It is great to see some direct field measurements of glacier thickness and bed topography, yet the relevance depends on the accurate definition of the reservoir area (see Specific comments). Author response: We will state the location of the figure. Changes in the manuscript: We will indicate the location of the GPR line in the figure caption.

Comment from Referee: Line 382. Consider the tittle "Temperature and precipitation trends", patterns would be more adequate for analysys of the geographic distribution of the variables Author response: Thank you for this comment. Changes in the manuscript: We will change the heading title to "Temperature and precipitation trends"

Comment from Referee: Line 395. This is a bit simplistic since ELA depends on both temperature and precipitation. Author response: Thank you for this comment Changes in the manuscript: We will remove lines 395-397 as they are speculative.

Comment from Referee: Line 402. Verify if the unit is m.e. instead of m2.
 Author response: This is a bit of an odd unit, but it should be correct given the calculations necessary to calculate the variance. Changes in the manuscript: We have verified the units.

Comment from Referee: Line 405. After enlarging consider including 5 yr markes on the x axis of Fig 7. Author response: Thank you for this comment. Changes in the manuscript: We will include 5 year ticks on the x axis of this figure.

REFEREE 2

Comment from Referee: A major issue is the question of whether the timing of surge onset is related to cumulative snow accumulation, as has been demonstrated at Var-
iegated Glacier by Eisen et al. (2001). A large proportion of the paper (pp. 133-200, 287-319, 417-444) is devoted to this question, using firn- and ice-core data to construct time series of annual accumulation and comparing them to the intervals between surge events. However, as the authors point out, at Donjek Glacier the surge reservoir zone is in the ablation area, âĽij15 km downstream of the snowline. Thus elevation changes are driven not by snow accumulation, but by the flux divergence minus surface ablation rates. It is clear that the flux divergence term is dominant in this case. During surge build-up, the ice thickens due to convergent flow (which more than offsets melt), while during the surge phase the ice thins faster than the melt rate due to divergent flow. There is no reason to expect that dynamic thickness changes in the ablation zone during surge build-up are correlated with annual accumulation rates some 30 km upstream, even when the latter are adjusted to account for transport time. Ice is not delivered to the reservoir zone one annual increment of snowfall at a time. The integrated mass balance upglacier of km21 will determine the balance flux, but the distances involved mean that the balance flux will be insensitive to annual variations in accumulation. Author response: The question of flux divergence and convergence is a very interesting one and merits further investigation. Although presently beyond the scope of this paper, we are able to highlight the areas of the reservoir and receiving zones. It is true that ice delivered to the reservoir zone does not come one year at a time, which is why we highlight the cumulative accumulation (e.g., Fig. 4), showing how much mass accumulates between surges at Eclipse Icefield. We focus on accumulation over the course of a quiescent phase, not annual accumulation, to highlight the potential relationship between the accumulation area of the glacier and the portion involved in the surge events. While part of the question is how does the mass move through the system (as the referee describes), the other part of the question is what causes the surge to initiate? Our data attempt to answer this question, and although more information would be useful, they are currently the best dataset we have to answer this question. This allows us to better understand the controls on the surge periodicity and its long-term evolution. The accumulation data are appropriate to answer this question

as the glacier geometry stays fairly constant through time, only changing as the glacier surface moves up and down due to mass flux. Changes in the manuscript: We will modify the text to better state our motives in understanding long-term cumulative accumulation and the controls on surge periodicity. We will add language to discuss the flux convergence and divergence as is described in our JoG paper, with appropriate citation. We will also reemphasize the focus on quiescent phase accumulation, not annual accumulation.

Comment from Referee: A more fruitful line of inquiry would be to examine time-series of elevation changes in the reservoir zone itself, and to relate these to dynamic cycles. At present, there is no discussion of the important observations by Abe et al. (2016), which revealed consistent velocity patterns over the last two surge cycles. What are the causal relationships between ice dynamics and elevation changes, and how do they evolve over the surge cycle? How & when do the dynamics influence elevation change (e.g. through cycles of divergent vs. convergent flow), and how & when do elevation changes influence the dynamics (e.g. via changing shear stresses or other factors)? The ice surface elevation change data presented in the paper are obviously too sparse to examine these issues in detail, but a good starting point would be to look at the trends in ice elevation shown in Fig 5 alongside the dynamic patterns revealed in Abe et al's velocity records (their Fig 1c and d). In particular, note how the patterns of elevation change correspond to the spatial patterns of velocity during quiescence (and, importantly, the variations in glacier width, particularly the 1/3 reduction in glacier width between km22 and km18). The narrowing of the glacier indicates that the glacier needs to speed up below km20 to satisfy continuity, but it either undershoots (quiescence) or overshoots (surge) the required value. Why should this be? Author response: These are excellent observations and ones we have considered carefully. We refer the referee to our recent paper in the Journal of Glaciology (https://doi.org/10.1017/jog.2019.34), which adds additional velocity measurements beyond Abe et al (2016) and highlights patterns we see on Donjek Glacier and how those change during the last three surge cycles. We also discuss the relationship between the velocity and elevation measurements there. We believe this paper is complementary and builds on our work presented here. Changes in the manuscript: We will better integrate the results and references from our recent JoG paper.

Comment from Referee: In a throwaway statement on line 510, we read that the upper extent of surge behavior is "coincident with a change in bedrock lithology". The fact that the dynamic instability occurs at both a topographic and geologic transition deserves to be investigated and reported in detail. What exactly is the lithological change? How does this relate to valley morphology? How might this affect the ability of the glacier to evacuate basal meltwater? If they hope to understand the surging behavior of Donjek Glacier, the authors need to give serious consideration to the idea that the instability relates to topography and/or geology and how it interacts with the dynamics. Knowledge of the subglacial topography would add a great deal of value in this respect, but the current sampling shown in Fig. 8 is far too sparse to allow any useful analysis. Author response: We completely agree and are working hard to get back to the field to collect more GPR data in the coming years. This is another paper in the making and is beyond the scope of this paper, which addresses the question of long-term controls on surge periodicity, rather than local controls on exactly where surges initiate and terminate at Donjek Glacier. Changes in the manuscript: We will incorporate a more through discussion of the bedrock change and associated glacier changes.

Comment from Referee: The subject of surge onset and weather shows promise, but this too will require more work. The authors state "three of the top ten rainiest months appear to coincide with surge onsets" (410). This of course means that the other seven rainiest months do not, but there could be something in this and it should be investigated in more detail. Perhaps rain events do have an influence, but only if the glacier is 'primed', or close to a critical state (i.e. if the surge front is close to the terminus, or some other condition is met). Can a particularly rainy month trigger a surge a year or two sooner than average? Conversely, can a dry summer delay surge onset causing the period to be longer than average? Detailed figures showing monthly precipitation

totals relative to surge onsets would help shed light on this (resolution of the monthly precipitation record in Fig 7d is currently too low to convey useful information). Author response: We too agree that a closer examination of the surge onset and precipitation is needed. However, the data currently available for precipitation are only from the Burwash Landing weather station and reanalysis. The weather station data have large data gaps, which make some of this work challenging. Moreover, the temporal resolution of surge onset is still coarse, particularly for older surges. More work therefore needs to be done to temporally constrain the surge onset, which can hopefully be done during the next surge event in the coming years. Changes in the manuscript: We will include a more careful discussion of the precipitation data and provide additional qualifiers to show these preliminary results, which merit further investigation.

Short comments from Mauri Pelto

Short comment from Mauri Pelto: 344: Suggested clarification and supporting references.... Transient snowline should be used instead of simply snowline. "Our remote sensing analysis illustrates that the summer transient snowline (TSL) in the center flow unit of Donjek Glacier has migrated up-glacier by 55 m yr-1 horizontally and risen by âĹij1.0 m yr-1 in elevation over the period 1951 to 2017 (Figures 6 and 7a). The increase in TSL elevation during this period has been observed on other glaciers in the region including Brady Glacier and Taku Glacier (Pelto et al, 2013a and 2013b). Over the study period the TSL was lowest in 1977 (Figure 7a), with an accumulation area of 337.3 km2 and an Accumulation Area Ratio (AAR) of 75.3%. The TSL reached its highest average elevation of âĹij2550 m a.s.l. in 2017, corresponding to an AAR of 68.4%. The higher TSL is indicative of a reduced surface mass balance." Author response: We appreciate this comment and will make the associated change. Changes in the manuscript: We will change the language as such: "Our remote sensing analysis illustrates that the summer transient snowline (TSL) in the center flow unit of Donjek Glacier has migrated up-glacier by 55 m yr-1 horizontally and risen by âĹij1.0 m yr-1 in elevation over the period 1951 to 2017 (Figures 6 and 7a). Over the study period

the TSL was lowest in 1977 (Figure 7a), with an accumulation area of 337.3 km2 and an Accumulation Area Ratio (AAR) of 75.3%. The TSL reached its highest average elevation of âĹij2550 m a.s.l. in 2017, corresponding to an AAR of 68.4%. The higher TSL is indicative of a reduced surface mass balance."

Short comment from Mauri Pelto: 349 The following statement needs supporting details to illustrate how Donjek Glacier is different based on the observation dates when the maximum TSL is achieved and if there is any trend in the timing of the TSL or any migration rate observations available, that would support the statement below. Author response: Thank you for pointing this out and ensuring that we back up our statements. Changes in the manuscript: We will show the monthly average snowline elevation for the entire dataset to back up our claim that the measurements are not biased by seasonal aliasing.

Short comment from Mauri Pelto: 539: It is noted that the timing of surge events has been consistent, but the size has continued to decline. Should it be added that "This decline in surge volume coincides with a rising snowline indicative of reduced mass balance." The reduced mass balance is evident regionally Das et al (2014) and Larsen et al (2015) as well as on Donjek Glacier. This could be observed at 465 too. Author response: We appreciate the suggestion of further emphasizing the change in mass balance and surge volume. Changes in the manuscript: We will add the sentence on line 540 suggested by Dr. Pelto: "This decline in surge volume coincides with a rising snowline indicative of reduced mass balance."

Please also note the supplement to this comment:
https://www.the-cryosphere-discuss.net/tc-2019-72/tc-2019-72-AC1-supplement.pdf